# Disease History, Pathogenesis, Diagnostics, and Therapeutics for Human Monkeypox Disease: A Comprehensive Review

**DOI:** 10.3390/vaccines10122091

**Published:** 2022-12-07

**Authors:** AbdulRahman A. Saied, Manish Dhawan, Asmaa A. Metwally, Mathumalar Loganathan Fahrni, Priyanka Choudhary, Om Prakash Choudhary

**Affiliations:** 1National Food Safety Authority (NFSA), Aswan Branch, Aswan 81511, Egypt; 2Ministry of Tourism and Antiquities, Aswan-office, Aswan 81511, Egypt; 3Department of Microbiology, Punjab Agricultural University, Ludhiana 141004, India; 4The Trafford Group of Colleges, Manchester WA14 5PQ, UK; 5Department of Surgery, Anesthesiology, and Radiology, Faculty of Veterinary Medicine, Aswan University, Aswan 81528, Egypt; 6Faculty of Pharmacy, Universiti Teknologi MARA (UiTM), Puncak Alam Campus, Selangor Branch, Puncak Alam 42300, Malaysia; 7Department of Veterinary Microbiology, College of Veterinary Science, Guru Angad Dev Veterinary and Animal Sciences University (GADVASU), Rampura Phul 151103, India; 8Department of Veterinary Anatomy and Histology, College of Veterinary Sciences and Animal Husbandry, Central Agricultural University, Selesih, Aizawl 796014, India

**Keywords:** monkeypox, outbreak, diagnostics, infectious diseases, history, pathogenesis

## Abstract

The monkeypox disease is a zoonotic-infectious disease that transmits between animals and humans. It is caused by a double-stranded DNA virus belonging to the *Orthopoxvirus* genus that is closely related to the variola virus –the causative agent of smallpox. Although monkeypox infections were endemic to Western and Central Africa, the newly emerging monkeypox outbreak spread to more than 90 non-African countries. With the exception of the PCR-confirmed case of a return from Nigeria to the United Kingdom, the ongoing outbreak is largely unrelated to travel. In the most recent wave, cases are characteristically males in their thirties. Risk factors include close and particularly sexual contact with an infected person, and contact with fomites, infected animals or aerosolized-infectious material. Clinical diagnosis of monkeypox is confirmed with nucleic-acid amplification testing of samples originating from vesicles or genital lesions and using real-time or conventional PCR. Other methods, such as electron microscopy, immunohistochemistry, and virus culture are costly and time-consuming. In addition to timely diagnosis and contact tracing, restrictive measures to limit spread, such as isolation of infected patients, preventing contact with wild animals, and isolation of animals suspected to be viral reservoirs have shown promise. Although there are no specific treatments for monkeypox disease, the experience with smallpox suggests that the vaccinia vaccine, cidofovir, tecovirimat, and vaccinia immune globulin (IVG) may be beneficial for monkeypox treatment. In this review, we provide an update on the human-monkeypox disease with a special emphasis on its pathogenesis, prevention, diagnostics, and therapeutic measures.

## 1. Background

Monkeypox (MPX) is a rare zoonotic disease caused by the Monkeypox virus (MPXV), a double-stranded DNA virus belonging to the *Orthopoxvirus* genus. MPX is endemic to Western and Central Africa. MPXV is closely related to the variola virus, the causative agent of smallpox. In the past 20 years, deforestation, population development, encroachment on animal habitats, more human migration, and increased global interconnectedness may have led to MPXV expansion to fill the ecological niche once occupied by the closely related smallpox virus [1,2,3]. The World Health Organization (WHO) deemed the risk to global public health posed by the multi-country monkeypox outbreak in non-endemic countries to be “moderate” on 29 May 2022 [4]. Since then, however, monkeypox is classified as a High Consequence Infectious Disease (HCID) in the UK (https://www.gov.uk/guidance/high-consequence-infectious-diseases-hcid, accessed on 7 November 2022). On 23 July 2022, WHO declared the ongoing global monkeypox outbreak a Public Health Emergency of International Concern (PHEIC), and it is the second time in two years that the WHO has taken the extraordinary step of declaring a global emergency. There is an urgent need for instant access to clear, succinct, fact-based information as monkeypox is mostly unknown to medical professionals especially to front-line healthcare personnel in prehospital, emergency departments, hospitals, and acute care/sexual transmitted illness clinics [5]. To distinguish the current outbreak from recurrent local cases in known enzootic regions, a new nomenclature has been proposed [6], with the Congo Basin lineage as clade 1, the West African lineage as clade 2, and the clade 2 (clade 2a and clade 2b) variants are the main ones circulating in the current global outbreak. The changing epidemiology of human monkeypox presents its own challenges. However, what is known is that immunocompromised patients are more susceptible to the disease. Because monkeypox resembles smallpox, antiretroviral therapy, although it has limitations for use in the former at present, may shed some light on decelerating disease progression, particularly in immunosuppressed individuals.

## 2. History of Monkeypox

The virologist, Preben Christian Alexander von Magnus, discovered and named monkeypox in 1958 in Denmark while investigating two smallpox-like disease epidemics that happened in laboratory monkey (*Cynomolgus*) colonies [7] whose phylogenetic data revealed that it was from the West-African Clade [8]. However, some researchers have proposed that MPXV might be evolved before 1958, the year of its discovery in Denmark [9,10]. In September 1970, a 9-month-old child, who was admitted for suspected smallpox to the Basankusu Hospital in the Republic of the Congo (nowadays known as the Democratic Republic of the Congo; DRC), is considered the first known human MPX case [11]. Then, human MPXV cases were discovered in Liberia, Sierra Leone, and Nigeria [12,13]. Numerous subsequent outbreaks of human monkeypox have often been recorded in Equatorial Africa, particularly in the DRC and Nigeria [14,15,16]. The Central-African (Congo Basin) and Western-African clades are the two clades that currently exist. In 2003, the first monkeypox outbreak outside of Africa originated in West Africa (Ghana) and occurred in the United States. All cases resulted from contact with sick prairie dogs without reported human-to-human transmission [17] (Table 1).

## 3. The 2017–2018 Outbreak in Nigeria

In the 2017–2018 outbreak in Nigeria, the case fatality rate was 6% with 122 confirmed or probable cases of human MPX recorded, including seven deaths. A total of ten patients reported contact with animals (two with monkeys, two with rodents, two with unspecified wild animals [consumed as meat—i.e., bush meat], and four with domestic animals). None reported contact with sick or dead animals. The patients reportedly presented with vesiculopustular rash, fever, pruritus, headache, and lymphadenopathy. The rash typically affected all parts of the body, with the face being the most affected [15]. Patients with monkeypox often have a febrile prodrome (4–17 days), which is characterized by lymphadenopathy. The prodrome is followed by the start of a deep-seated, vesicular, or pustular skin rash with a centrifugal distribution 1–4 days later. Some recent cases have manifested atypically, without the subjective temperature or other prodromal symptoms. As a result, cases may be mistaken for more prevalent viruses, such as varicella-zoster or sexually transmitted infections (STIs) (e.g., genital herpes or syphilis). In the most recent global outbreak, the virus can transmit through direct contact with infectious sores, scabs, or body fluids, and shared beddings or clothing. More strikingly, in the current outbreak, cases have been atypical, with the characteristic rash starting in the genital and perianal areas with or without dissemination to other parts of the body [32]. Patients are considered infectious once the prodroma or rashes begin, and until lesions or scabs fall off. The distribution of cases and contacts suggested both primary zoonotic and secondary human-to-human transmission. Among prison inmates, the genomic analysis suggested multiple introductions of the virus and a single introduction, along with human-to-human transmission. Two nosocomial infections among healthcare workers were also reported [15]. The higher number of infections discovered in Nigeria has revealed that fatality, serious sickness, and human-to-human transmission are all possible outcomes [33]. The clinical outcome of the monkeypox infection in Nigeria was influenced by HIV status and presence of coinfection [34]. The incidence of MPXV in Nigeria in 2017 was more prevalent in males than females (ratio of 3:1). Strikingly, active surveillance data from the 1980s (0.72 per 10,000) and 2006–2007 (14.42 per 10,000) in the same health zone in DRC suggested a 20-fold rise in human monkeypox incidence [35] and vaccinated persons had a 5.2-fold lower risk of monkeypox than unvaccinated persons (0.78 vs. 4.05 per 10,000) [35]. Before the 2017 Nigeria outbreak, the majority of human-monkeypox cases were found in rural, forested areas of Africa; however, monkeypox cases have been found in urban areas; some of them have been severe, even resulting in death, most commonly in persons with HIV infection, implying new risk factors [15]. Over the past five years, hundreds of cases of monkeypox have been reported in Nigeria, with many cases among men, some of who had genital lesions, suggesting human-to-human transmission via sexual contact [33]. After 2017, the virus was spreading in an unfamiliar way: it was appearing in urban settings, and infected people sometimes had genital lesions, suggesting that the virus might spread through sexual contact [15].

## 4. The 2022 MPXV Outbreak in Africa

Without any history of travel to any of the nations where outbreaks occurred, South Africa reported its first case of the human-monkeypox. Among the western and central African countries, Nigeria, Cameroon, and the Democratic Republic of the Congo are namely a few of struggling with the MPX epidemic (Table 2). It is vital that discussions on vaccine access, testing, and antiviral therapies include these countries. 

This is a valid interpretation of what has happened with MPX, which has plagued countries in Africa for decades; yet, there was a lack of attention and research funding to tackle it. Before the Ebola outbreak in West Africa, Ebola-virus disease suffered the same fate; it was an obscure disease that nobody cared about, primarily causing deadly outbreaks in the Democratic Republic of Congo. When the western African outbreak was declared a PHEIC in July 2014 [36] (albeit after a much-criticized four-month delay by WHO), it moved forward more than 10 years of stalled progress on vaccines and therapeutics research for the disease, which had been neglected for years, putting the world in a better position to respond to outbreaks that have come since. Democratic Republic of Congo and Nigeria are the most affected countries. Rimoin et al. reported a 20-fold increase in the number of cases in the Democratic Republic of Congo between the 1980s and mid-2000s [35]. The estimated mortality of the west African clade of MPXV responsible for the ongoing outbreak (based on previous outbreaks) is around 3–6% [4].

As vaccines are deployed globally, African researchers worry they will be left behind [37]. More than 31 million doses of smallpox vaccine have been pledged from member nations of the WHO for smallpox emergencies, but these have never been sent to Africa for use against monkeypox [37]. The WHO has collaborated with African nations that are experiencing outbreaks of monkeypox to strengthen monitoring and diagnoses [37]. The goal for renaming the monkeypox-viral strains is to lessen stigma [37] and to discriminate nomenclature typically associated with geographic regions, nations, economies, and people [6].

The current outbreak differs from earlier ones in terms of age (most of those affected are in their thirties), sex/gender (most cases are male), risk factors, and mode of transmission, with sexual transmission being highly likely. Along with being characterized by anogenital lesions and rashes that largely spare the face and limbs, the clinical appearance is also atypical and distinctive [38]. Due to the atypical disease presentation and transmission patterns, the recent MPXV surge is “rare and unusual”, compared to earlier outbreaks, showing a significant shift in the epidemiological trend of MPX. MPX was largely a disease of young children in the early years (1970–1989), with a median age of 4–5 years when the clinical presentation was exhibited. The median age increased to 21 years within a span of 10 years, from 2000–2009 to 2010–2019. However, in the US outbreak in 2003, 10 out of 34 (29.41%) participants were under the age of 18 [2]. There were a few probable risk factors identified, such as being a young male, having sex with other men, engaging in risky behaviors and activities, including unprotected sex, human immunodeficiency- virus positivity, a history of previous sexually transmitted infections, including syphilis, and vertical transmission to an unborn child [38].

## 5. The 2022 Monkeypox Outbreak Outside Africa

On 18 May 2022, Portugal, Spain, and Canada reported 14, seven, and 13 cases of MPX infection, respectively [39]. Additionally, many other European countries, such as Belgium, Sweden, and Italy, announced their initial MPX instances on 19 May 2022. Australia identified two incidents on 20 May 2022 [40]. The individuals just returned from a trip to Europe. On 20 May, the Netherlands, Germany, and France all reported their initial cases. On the same day, UK’s Health Secretary revealed an additional 11 monkeypox cases, bringing the sum above 70 cases [40,41]. As of 23 November 2022, around 80,000 laboratory-confirmed monkeypox cases (https://www.cdc.gov/poxvirus/monkeypox/response/2022/world-map.html, accessed on 13 November 2022) have emerged in non-African countries with over 40 confirmed deaths since the first confirmed case in the UK on 7 May [42]. Although the first case of the UK was travel-linked to Nigeria, there are several unusual cases with no travel history and nor linked with previous cases as seen in Gay Pride parties (where sexual activities took place) in the Canary Islands, rave events in Berlin, Germany, and Madrid, Spain [43,44]. In addition, the first case of monkeypox not linked to travel to endemic countries was reported in the United Kingdom (UK) on 13 May 2022 [45].

Since then, incidents have been documented in numerous Western countries, mostly among men who have sex with men (MSM) seeking care in primary care and sexual-health clinics for treatment [46]. MPX is usually self-limiting, but concerns are being raised about the existence of the virus in non-endemic populations with untrained immunity, wherein MPXV has not been identified before. In Europe or the US where high-resource settings, MPXV has taken a very different epidemiological landscape. People became worried about the atypical emergence of the 2022 MPXV outbreak in countries where the disease has never been reported, and in people who had never traveled to endemic countries. A probable under-detected community transmission was hypothesized [47].

The duration of the incubation period has been reported to differ by route of transmission for monkeypox virus, smallpox, and vaccinia viruses [48]. Around 90% of Portuguese cases reported sexual exposure [49]. The majority of MPXV cases had genital or peri-genital lesions, suggesting a new route of disease transmission that involves sexual contact. Health professionals from all around the world have proposed various possibilities that are currently being investigated, a definitive cause for the rapid spike in cases has not yet been identified. As many MPVX cases involve MSM, a deeper understanding of viral transmission will allow the development of a sex-based approach to disease screening and treatment [50]. The human-to-human transmission was reported on a small scale in endemic countries [51]. Nosocomial transmission had also been reported [52]. It was unclear about whether or not MPXV can spread through the air. Nevertheless, we cannot rule out an airborne transmission [53].

A zoonotic transmission occurs through direct contact with blood and bodily fluids, inoculation via mucocutaneous lesions of an infected animal, as well as through direct contact with or consumption of one of the natural viral hosts [52] (Figure 1). Strikingly, during the most recent outbreak, the first case of monkeypox inter-species transmission occurred between a human and a dog [54]. Infected human, animals, and contaminated objects transmit disease via direct contact and viruses are also known to cross the placental barrier [42,55]. In addition, the MPXV was detected in the seminal fluid taken from several of the cases examined [38]. De Baetselier et al. [56] reported that MPXV can transmit from an asymptomatic patient to close contacts, as shown among asymptomatic-monkeypox cases. Therefore, contact tracing is rather challenging and impractical, particularly when it involves numerous anonymous partners [38] (Figure 1). Nevertheless, contact tracing is crucial in controlling the spread of monkeypox.

## 6. Pathogenesis and Infection Mechanism

The nasopharynx, or oropharynx, along with subcutaneous portals, can be exploited by the MPXV to infect its host. MPXV multiplies at the entry site before spreading to nearby lymph nodes. The virus spreads to certain other organ systems after an early phase of virus infection. MPXV resembles other-recognized *Orthopoxviruses* in terms of appearance. The outer membrane of MPXVs, which are oval or brick-shaped, is made of lipoproteins [1]. Despite being a DNA virus, the MPXV completes its life cycle in the cytoplasm. During viral DNA replication, transcription, and virion packaging, a number of proteins are necessary [57]. MPXV can enter or penetrate the host cell through fusion and macropinocytosis [58,59]. The occurrence of two separate viral types, intracellular mature virus (IMV) and extracellular enveloped virus (EEV), which are enclosed by various different lipid membranes and also exhibit specific surface proteins, complicates the entrance and release of MPXV [60] (Figure 2). Additionally, the Congo clade of the MPXV might differ from the West African clade as the Congo clade exhibit more virulence and pathogenicity [61].

After the DNA replication, transcription, and translation events in the host cytoplasm, IMVs are formed within the cytoplasm. Additionally, enclosed viruses can finish viral assembly by budding through the plasma membrane in the form of EEV. IMV becomes EEV via encapsulation by intracellular membranes and is carried to the cell surface on microtubules and exposed on the cell surface via exocytosis, or they are released upon cell lysis (Figure 2). Furthermore, the enveloped virion is known as a cell-associated enveloped virus (CEV) if it stays affixed to the cell surface. CEV spreads into neighboring cells by developing actin tails below the plasma membrane. A different option is to release the surface virion as EEV [60,62,63], which can further infect the neighboring cells.

According to the serological analysis of cytokine responses to human MPXV infection, human-monkeypox illness is thought to be accompanied by a cytokine storm. Following MPXV infection, researchers also discovered indications of a strong T helper 2 (Th2) response and a weakened Th1 response. IL-4 (and the related IL-13), IL-5, and IL-6 levels were increased above the normal human range, while IL-10 levels were high in severe instances. These cytokines are linked with Th2-mediated immune response. Tumor necrosis factor-alpha (TNF-alpha), interferon-alpha (IFN-alpha), interferon-gamma (IFN-gamma), and IL-2 were shown to have lower serum concentrations [64]. Hence, the dysregulation in the Th1-mediated immune response can be associated with the severity of the infection. It is interesting to note that the generation of inflammatory cytokines is suppressed in the human cells that have already contracted the monkeypox virus.

## 7. Clinical Features and Risks for Infection

Following exposure and infection, there is an incubation phase of around 10 to 14 days, followed by a prodrome period of approximately two days. An infected individual may have a high fever, cold, fatigue, headache, sore throat, breathlessness, and enlarged lymph nodes during the prodrome phase (the time before the appearance of a rash) [3,65]. It is important to note that the presence of swollen lymph nodes in the submandibular, cervical, or inguinal areas may serve as a marker for separating human monkeypox from human-smallpox infection [66]. About 90% of all cases of MPX infection display it [65,66].

A gradual maculopapular rash with lesion diameters ranging from 0.2–1 cm appears after the prodrome stage [3,65]. The diseased individual is also thought to be the most infectious during this period [65]. Initially on the face and neck, and moving to the legs with involvement of the palms and soles, lesions spread throughout the body in a centrifugal manner. From macules through papules, vesicles, pustules, and ultimately a crusting phase that involves scabbing and desquamation, lesions advance through many phases during a two- to the four-week period [67]. Lesions have sometimes left behind dyspigmented scars. Extracutaneous symptoms, including pneumonitis, eye problems, secondary skin and/or soft tissue infection, and encephalitis, have sometimes been reported [3,66].

Nevertheless, monkeypox infection’s clinical manifestations are similar to a milder case of smallpox. The distinction is that lymphadenopathy is brought on by MPXV infection rather than smallpox. Fever, chills, headache, muscular pains, backaches, and weariness are the first symptoms of an MPXV infection, which proceed to exhaustion [63,68]. Monkeypox infection Mon typically takes seven to 14 days to incubate, but it can require up to 21 days. The infection spreads to other body parts after the onset of a fever and the development of a rash on the face [69]. The oropharynx is where lesions first form before spreading throughout the body. Nearly two weeks after exposure, serum antibodies can be detected. Depending on the lineage of the pathogenic-viral strain of MPX and the accessibility of contemporary healthcare, the death rate can vary from 1 to 10% [63,70].

Unquestionably, the tendency in non-war areas for many people to gather after pandemic-related curfew and home isolation increase the likelihood of getting the infection in vulnerable groups [50,71,72,73]. Spontaneous early miscarriages and a second-trimester loss at 18 weeks’ gestation were recorded in pregnant women in the Democratic Republic of the Congo between 2007 and 2011 [55]. MPXV DNA was found in fetal tissues, the umbilical cord, and the placenta, proving that the monkeypox virus is transmitted vertically [55]. Even though there are no obvious epidemiological links, clinicians must retain a high index of suspicion for the monkeypox virus in any pregnant woman presenting with lymphadenopathy and vesiculopustular rash, particularly rash localized to the vaginal or perianal region. Fetal ultrasound could be required for testing asymptomatic pregnant women with significant monkeypox-virus exposure [72].

## 8. Diagnostic Management

If a person exhibits the aforementioned symptoms, monkeypox may be suspected, especially if that person has a history of contact or travel to monkeypox-endemic regions. Once clinical suspicion exists, diagnostic evaluation of monkeypox should be initiated (Figure 3). Importantly, diagnosis of the monkeypox during the current outbreak differs from what is mentioned in the classic descriptions in the West African region due to atypical transmission, sustained human-to-human transmission, and history of sexual contact. Therefore, even if the rash is localized and not (yet) extensive, monkeypox should be considered in the differential diagnosis when a patient appears with STI-associated or STI-like rash [74] and also when evaluating genital-ulcer diseases [75]. Monkeypox and varicella (chickenpox) are sometimes mistaken in countries where these infections are prevalent.

Based on clinical diagnosis, the detection of monkeypox may be misdiagnosed with some sexually transmitted infections (STIs), such as syphilis, herpes simplex, and disseminated-gonococcal infection. Further, monkeypox is most easily recognized by its propensity to cause moderate to severe lymphadenopathy. Typically, monkeypox cannot be clinically distinguished from other pox-like viruses, such as varicella, herpes zoster, measles, and arboviruses (Dengue, Zika and chikungunya), making a diagnosis based only on clinical observation difficult and insufficient. Therefore, it could not be used as a fast-track infection control strategy. Hence, nucleic-acid-amplification testing for the monkeypox virus from vesicles or genital lesions using real-time or conventional real-time polymerase chain reaction (RT-PCR) is used to confirm the diagnosis because of its high accuracy and sensitivity approaching 100%.

The diagnosis of human MPX is made on the basis of a clinical suspicion backed by typical skin and mucosal lesions, which is then verified by molecular testing [76]. The preferred strategy for diagnosing an active monkeypox case is the identification of viral DNA in swabs taken from crusts of vesicles or ulcers [77]. The RT-PCR assays target various *Orthopoxvirus* genes. DNA polymerase (E9L) and envelope protein (B6R), two distinct viral gene targets used in combination, showed 100% specificity for monkeypox in RT-PCR assays. This finding suggests that using two distinct viral-gene targets in combination could provide a reliable and sensitive method for rapid diagnosis [78].

Numerous types of skin lesions, including as macular, pustular, vesicular, and crusted lesions, as well as lesions in various stages that manifest at the same time, have been linked to MPX [79,80]. For monkeypox diagnostic tests, skin lesions, pustule and vesicle fluid, and dry crusts are the best diagnostic samples. Sterile-dry polyester, nylon, or Dacron swabs can be used to acquire two swab specimens from a minimum of three lesions. In samples of lesion crusts, dry lesion swabs in viral-transport media (VTM) are allowed [81].

The Center for Disease Control and Prevention (CDC) advises clinicians to obtain two specimens for each patient to make an accurate diagnosis. Each sample should be collected from multiple lesions, ideally from several different body sites [82]. The FDA advises clinicians to swab the lesion since blood and saliva could produce inaccurate results [83] because monkeypox virus remains in blood for only a short period of time. Samples of urine, semen, rectal fluid, and/or genital tissue may also be collected, depending on the clinic.

Within an hour after collection, specimens should be refrigerated (2–8 °C) or frozen (−20 °C or lower), and they should be brought as quickly as possible to the lab. Samples should be stored for a longer period of time at 70 °C and repeated freeze-thaw cycles should be avoided. When a cold chain is not readily available, viral DNA from skin-lesion material can be considered since it is relatively stable when stored in a dark, cool environment [84]. For the time being, testing for monkeypox continues to be heavily concentrated in facilities with qualified staff mainly handling the specimens in Biosafety Level 2 (BSL-2) facilities.

Diagnosis of monkeypox starts with suspicion in cases suffering from pox-like lesions and a history of contact or travel to monkeypox-endemic regions. However, PCR is the preferred method to confirm the diagnosis. Whole-genome sequencing (WGS) is the gold standard for characterizing the monkeypox virus, but its use is restricted, particularly in developing countries. Low blood urea nitrogen (BUN), increased transaminase levels, hypoalbuminemia, leukocytosis, and thrombocytopenia are some of the laboratory findings reported in monkeypox. The methods of serology and antigen detection are not advised because it is insufficiently specific due to the significant cross-reactivity with other *Orthopoxviruses* [85,86]. Additionally, immunohistochemical (IHC) staining for *Orthopoxvirus* antigens and visualization on electron microscopy (EM) are among diagnostic methods [87]; however, they are rarely used in clinical practice due to high technical skills and facility requirements. Monkeypox culture-based testing should not be done frequently in diagnostic or clinical laboratories [82].

Additional TaqMan probe-based RT-PCR assays have been described as a generic monkeypox assay that targets both monkeypox clades [88]. But its access remains limited. Li et al. [89] developed Cepheid GeneXpert (a self-contained cartridge) to provide an alternative to traditional PCR detection methods. Regardless of the type of specimen obtained (crust versus vesicular swab), the GeneXpert assay demonstrated high sensitivity, specificity, negative predictive value, and positive predictive value in suspicious specimens [89], but it requires high costs. Chelsky et al. [90] validated a direct RT-PCR protocol for monkeypox viral identification to increase the scalability of monkeypox testing. The test eliminates the need for nucleic acid extraction kits, cuts down on lab tech time per sample, and reduces exposure to an infectious agent while maintaining the sensitivity and accuracy of the indirect assay.

The gold standard for characterizing monkeypox virus and other *Orthopoxviruses* is the next-generation WGS methods. Because of its high price and cutting-edge technology, its use is restricted, particularly in developing countries. 

Quality services to the public can be provided with documented guidelines for collecting and forwarding biological materials. Integrating monkeypox screening into routine surveillance systems is now indispensable. Varicella, herpes simplex, and syphilis could lead to misdiagnosis because these conditions can mimic monkeypox in pregnant women. Additionally, there is a need for developing diagnostics home-based test kits.

## 9. Preventive Measures

To limit the spread of the MPXV, several precautions can be implemented. Reducing direct interaction with animals believed to be the reservoir of the viral agent, for example, has been reported lately to be effective, particularly in areas where monkeypox cases were on the rise [91,92]. Isolating and euthanizing the animals suspected of being the virus reservoirs can also be an efficient approach to reduce the spread of the viral agent. Additionally, isolating infected patients in a room with negative pressure can stop the virus from spreading from person to person and thus reduce community transmission [63,93] (Figure 4).

Furthermore, it is essential to avoid direct contact with any objects that have come into contact with animals or people who were ill. The proper personal protective equipment (PPE), such as N-95 masks and fully covered water-resistant gowns which can prevent air-borne infectious agents, should be worn by front-line staff providing care to the viral-infected patients as well as high-risk people who are anticipated to come into contact with the infected persons [63].

Furthermore, Parvin et al. (2022) [94] postulated several biosafety measures to reduce person-to-person transmission, including: 1. immediate isolation of sick or potentially sick patients. 2. One can assist individuals in recovering from infections and improving their comfort by avoiding contacting oral and ocular rashes, allowing other rashes to dry, not covering the wound, or hydrating the rash with a damp cloth. Avoiding direct bodily contact, the crust of pus from rashes, and droplet contact from an infected person’s cough or sneeze can assist in prevention of the disease. 3. The viral infection might be stopped from spreading by using protective clothing (e.g., safety gloves, goggles, and masks), cleaning bedding and garments separately, properly discarding sanitary products, and isolating patients in a hospital or with their families. 4. It is strongly advised to implement surveillance and spread control programmes in high-risk nations in order to quickly identify and address any suspected MPX cases. 5. Community involvement and prompt alerting of health departments to suspected MPXV cases are needed 6. Public awareness of the virus and its modes of transmission needs to be raised 7. The One-Health strategy is put into practice during preparation and reaction [94].

## 10. Therapeutics and Vaccination

The mainstay of clinical management for typical a MPX infection is supportive and symptomatic treatment [95]. Limiting non-steroidal anti-inflammatory drugs (NSAIDs) is reasonable due to the concern of developing hemorrhagic lesions [96]. Currently, there are no specific treatments for monkeypox disease; however, experience with smallpox suggests that the vaccinia vaccine, cidofovir, tecovirimat, and vaccinia immune globulin (IVG) may have a use in monkeypox treatment [97]. As per the WHO guidelines, tecovirimat was developed for smallpox and was licensed by the European Medicines Agency (EMA) for monkeypox in 2022. Current antivirals, tecovirimat, brincidofovir, cidofovir, and vaccinia immune globulin intravenous (VIGIV), which are approved for smallpox infection, can also be used for monkeypox treatment, which could potentially outweigh their harm to those with severe diseases or poor prognosis, such as immunocompromised patients, pediatrics, pregnant and breastfeeding women, complicated lesions, and when lesions appear near the mouth, eyes, and genitals [98]. Stability testing and safe- dosing studies for these antiretrovirals have been conducted in humans, but evidence for their efficacy is limited [98].

A clean fluid composed up of highly concentrated IgG antibody against the vaccinia virus that were obtained from healthy individuals who had previously received a vaccination against the live vaccinia virus is known as vaccine immune globulin (VIG) [99]. The CDC permits the use of VIG to treat an epidemic of MPX. The treatment of *Orthopoxvirus* infection with VIGIV was documented in many investigations [100,101]. However, there is no information on how well VIG works to treat MPXV infection. In serious situations, doctors could think about utilizing it. In those cases with a severe impairment in T cell functioning, for whom smallpox immunization is contraindicated following exposure to MPXV, the VIG is also appropriate for preventive usage [102].

The CDC-held Emergency Access Investigational New Protocol allows the use of tecovirimat for non-variola *Orthopoxvirus* infections, such as MPX. The protocol also includes an allowance for opening an oral capsule and mixing its content with liquid or soft food for pediatric patients weighing less than 13 kg. Tecovirimat (orally, 600 mg BD for 2 weeks) is available through the Strategic National Stockpile as an oral-capsule formulation or an intravenous vial. Tecovirimat (formerly ST-246, with the trade name TPOXX^®^) inhibits the highly conserved p37 *Orthopoxvirus* protein responsible for the development of virions, inhibiting the spread of the virus within an infected host [103]. It is indicated for the treatment of smallpox in adults and pediatric patients weighing at least 3 kg. Tecovirimat has demonstrated efficacy against smallpox in both human and animal models [104]. Despite the fact that its effectiveness against monkeypox in humans has not been proven, investigations on animals treated with tecovirimat at various phases of the disease have shown better survival from lethal monkeypox-virus infections, compared to animals treated with placebo [104,105]. Tecovirimat was initially authorized in oral form in 2018; an IV formulation was authorized in 2022. Dual therapy may be used in severe diseases where brincidofovir is added. Both tecovirimat and brincidofovir are available in the Strategic National Stockpile. The European Medicines Agency has approved tecovirimat for monkeypox. Tecovirimat and vaccinia-immune globulin can be considered for treating pregnant women who are severely ill [72].

Cidofovir, the nucleotide analog, can inhibit the progression of both smallpox and monkeypox [106]. As of 6 June 2022, the US FDA has not given cidofovir approval to cure monkeypox. Since there is an outbreak, cidofovir may be administered under the proper regulatory authority, such as an IND procedure or an emergency use authorization (EUA). Intravenous-normal saline and probenicid therapy must be given concurrently with cidofovir. Brincidofovir has been approved for the treatment of smallpox in the US since June 2021. Brincidofovir (oral) is an analog of the intravenous-drug cidofovir, and may have an improved safety profile, namely less renal toxicity, compared to cidofovir [107]. These drugs work by inhibiting the viral DNA polymerase [108].

Based on the clinical presentation and other diagnostic information, clinicians should weigh the risks and advantages of starting a particular treatment. Compared with adults, paediatric patients suffer from more severe diseases and higher mortality. Challenges for managing them are unique and warrant a careful multidisciplinary approach. Safety and dosing of the different antiviral agents are still not confirmed for paediatric patients [109]. Because Tecovirimat has no embryotoxic and teratogenic effects detected in animal studies. Additionally, the US Centers for Disease Control and Prevention allow the use of the live-smallpox vaccination ACAM2000 in emergency situations, which provides 85% cross-protective immunity against monkeypox. However, ACAM2000 has a rare risk represented in preterm delivery, stillbirth, neonatal death, and potential adverse maternal reactions. The third-generation smallpox vaccination MVA-BN, which was recently licensed in the USA, Canada, and the EU, may be safer because it contains a virus that cannot replicate and has not been shown to cause problems during pregnancy [72].

The smallpox vaccine has been reported to provide 85% cross protection against Monkeypox. Moreover, in the 1980s, contact tracing of monkeypox cases revealed an overall attack rate of 7.2% and 0.9% for contacts unvaccinated versus those vaccinated, respectively. Additionally, ring vaccination i.e., vaccinating close contacts of confirmed cases can be successful if infectious cases are promptly diagnosed. The benefits of ring vaccination are double-fold: it can break the chain of virus transmission and bring to a halt the course of a severe disease [110]. With fading immunity among populations and non-availability of first generation smallpox vaccine for the public, newer generation vaccines have been developed. ACAM2000 and MVA-BN (modified vaccinia Ankara-Bavarian Nordic) are two vaccines licensed for use in the USA. MVA-BN is specifically developed for Monkeypox and is a live, non-replicating vaccine based on modified vaccinia Ankara, a live attenuated form of vaccinia virus. It is available in the US as Jynneos, in Europe under the brand name Imvanex, and Imvamune in Canada [111]. Modified vaccinia Ankara (MVA) is available as a third-generation smallpox vaccine [112]. MVA-BN (IMVANEX/IMVAMUNE/JYNNEOSTM) stain has been licensed in multiple countries for protection from smallpox or monkeypox. In non-human primate studies, two doses of MVA-BN provided 100% protection against a lethal challenge of aerosolized monkeypox [113]. MVA-BN^®^ can also be considered for post-exposure prophylaxis in selected patients, ideally within 4 days of high-grade exposure [114]. As live vaccines are contraindicated in immunocompromised patients, vaccinia immunoglobulin may be considered, although data on its effectiveness are unclear [65]. The CDC recommends vaccination within four days of exposure to prevent disease or up to 14 days after exposure to reduce the severity of the disease. More than 36,000 doses of the JYNNEOS vaccine are currently in the US Strategic National Stockpile. Although the CDC has recommended that the JYNNEOS vaccine be offered to close contacts of patients with monkeypox, this vaccine is currently not easy to access.

In the USA, the currently approved vaccines include ACAM2000, the purified clone of the Dryvax vaccine by Sanofi, and the modified vaccinia Ankara by Bavarian Nordic (JYNNEOSTM in the USA, IMVANEXTM in Europe, IMVAMUNETM in Canada). JYNNEOSTM is FDA-approved for both smallpox and monkeypox, and the USA Strategic National Stockpile contains both JYNNEOSTM and ACAM2000. The immunocompromised and those with atopic dermatitis should not be vaccinated with a second-generation vaccine, such as ACAM2000, but can receive a third-generation vaccine, such as JYNNEOSTM/IMVANEX/IMVAMUNE.

## 11. Challenges

Over the years, global MPX surveillance data and reporting systems have faced several challenges, such as asymptomatic cases of monkeypox that go undiagnosed, unskilled staff, poor laboratory support of infectious diseases countries where MPX was declared endemic, underreporting due to inadequate access to healthcare facilities in developing countries, lack of global coordination, and countries’ unwillingness to give full reports due to the economic and political consequences of such. To properly quantify the public-health burden and devise methods to reduce the danger of infections spreading further, improved surveillance, epidemiological analysis, and equitable access to therapeutics and vaccines particularly to low-middle countries are required [115]. Failing to recognize the significance of bringing the spread to a halt can have dire consequences, including loss of tourism and possible travel restrictions to affected countries. Concurrently, the public-health priority should be to protect front-line healthcare workers.

## 12. Conclusions

In comparison to the 2017–2018 plague, the changing epidemiology of the May 2022 human monkeypox outbreak presents challenges of its own. A public-health crisis is imminent should there be a delay in actions taken at the initial phases of the resurgence. Lessons were learned from past experiences: delaying treatment under the pretext that monkeypox is a self-limiting illness may not have been the best approach. As we now know more about the antivirals’ and the current availability of those which were used for smallpox, proactive measures can be taken to curb the disease spread. In addition to the measures taken to contain the reinfection rates and viability of the virus, post-exposure vaccination, antivirals, and immunoglobulins can be considered for patients, particularly for immunocompromised individuals. To say we are in dire need of the development of novel antiviral therapies based on knowledge of viral dynamics, balanced with drug toxicities and drug-induced adverse events, is an understatement.

An integrated and concerted package of public-health interventions should be implemented. Belgium was the first country to enact a 21-day obligatory quarantine due to monkeypox. Active disease surveillance, contact tracking, timely diagnosis, and the isolation or quarantine of close contacts are important public-health interventions to stop the spread of infection. A consistent definition of monkeypox that may update earlier working definitions and incorporate rare or unusual clinical manifestations of monkeypox lesions should be developed. Rapid communication of data and the implementation of public-interventional measures in healthcare systems are crucial for controlling the spread of MPX infections.

Misdiagnosis is another challenge: MPX could easily be mistaken for other dermatological diagnoses within sexual-health clinics or primary care (e.g., chickenpox, varicella zoster, herpes simplex, syphilitic chancre, gonorrhea, or molluscum contagiosum). To reduce human-to-human transmission, it may be necessary to limit close contact with people infected with MPX, including sexual contact. Healthcare workers are at the highest risk, and it is important to identify cases as early as possible. PPE should be made mandatory for healthcare workers dealing with such cases. Wearing masks would be an effective method to prevent the transmission via respiratory droplets. Careful handling of samples is extremely important. Vaccines should be stockpiled for people at the highest risk. Health screening and quarantine for individuals travelling from affected countries should be made mandatory. A quarantine period of three weeks has been suggested as of November 2022.

## Figures and Tables

**Figure 1 vaccines-10-02091-f001:**
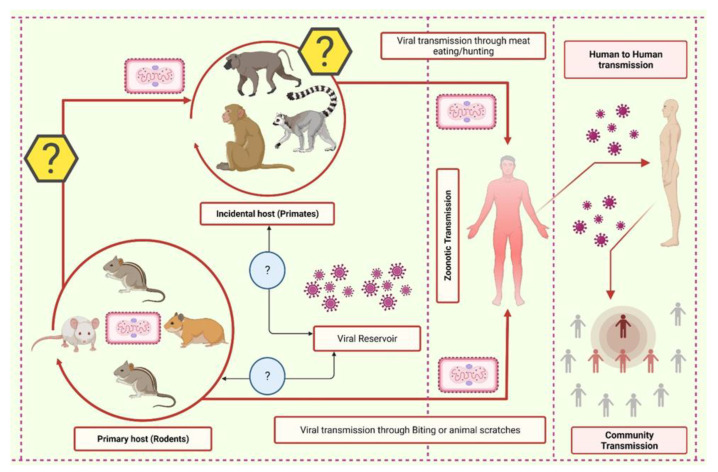
The schematic representation of the transmission cycle of the monkeypox virus. In wildlife, the virus rotates in the primary host and can be transferred to humans through various means. From the primary host, the virus can jump into the incidental host, which in turn can infect the human. Additionally, the reservoir of monkeypox is yet to be elucidated.

**Figure 2 vaccines-10-02091-f002:**
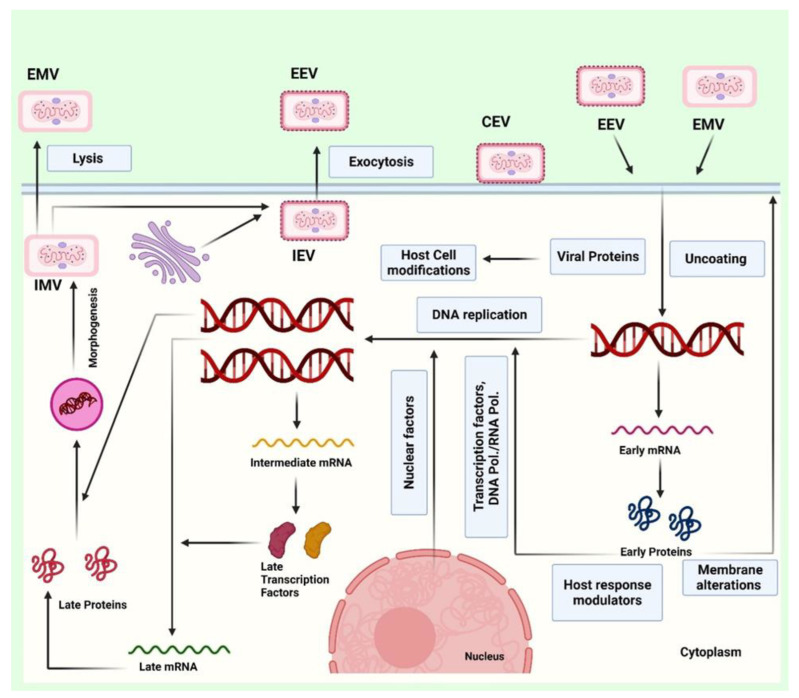
The figure represents the infection mechanism by which extracellular mature virion (EMV) or extracellular enveloped virus (EEV) infect the human cell and hijack the host machinery. The viral particles enter or penetrate the host cell through fusion and macropinocytosis, then replicate and infect the host cells. The genetic material replicates in the host cell and transcribes to produce the viral proteins with the help of host translation proteins. The genetic material and the viral proteins assemble in the host cell and are lysed through exocytosis. The assembled viral particles excytosed in the form of EMV and EEV.

**Figure 3 vaccines-10-02091-f003:**
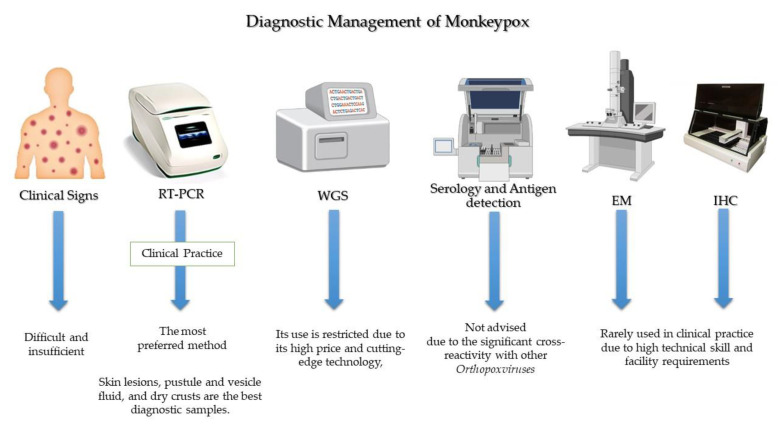
General lines for diagnosis of monkeypox.

**Figure 4 vaccines-10-02091-f004:**
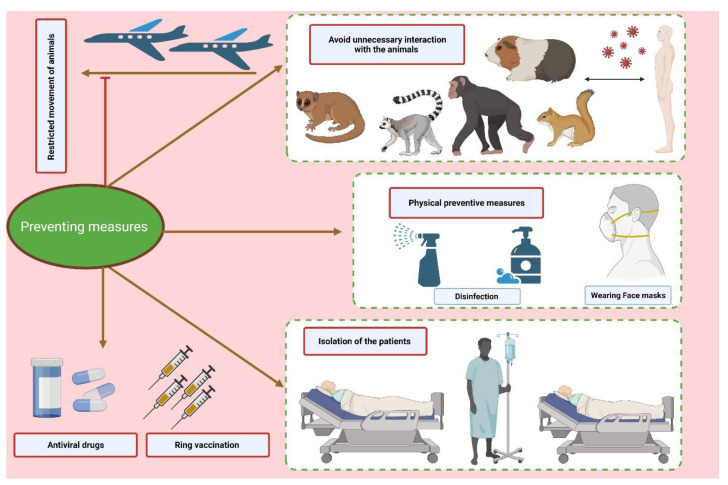
The figure represents the various preventive measures to contain the deleterious consequences of the monkeypox outbreak.

**Table 1 vaccines-10-02091-t001:** Timeline of Monkeypox virus-related events.

Year	Event	References
3500 years ago	Separation of MPXV within Old World *Orthopoxviruses* took place.	[18]
600 years ago	MPXV West African subtype appeared	[18,19]
1899	Identification of MPXV in five species of African rope squirrel (*Funisciurus* sp.) collected across Central Africa.	[20]
1958	MPXV was first identified in Denmark following an outbreak of the pustular disease in a macaque colony (*cynomolgus monkeys*). The macaques had been imported from Singapore.	[7]
1959–1964	MPXV, which was reported among colonies of captive monkeys, were also described in the USA (1959 and 1962) and Rotterdam Zoo, the Netherlands (1964).	[21]
1970 1972	The first human monkeypox case was identified in the Democratic Republic of Congo (DRC, then Zaire).	[11]
1970–1979	A reported outbreak of human MPXV in Nigeria.	[1,22]
1980	Vaccination laid the basis for the eradication of Variola (genetically related to MPXV)	[23]
1996–1997	A major outbreak of human monkeypox occurred in Katako-Combe, Zaire (DRC). A total of 73% of cases reported contact with another human case while 27% had known contact with a wild animal.	[24]
2003	An outbreak of human monkeypox occurred in the USA (more than 71 infected people). It was initiated by rodents (small mammals) imported from Ghana to be sold as exotic pets and have been transmitted by pet prairie (*Cynomys* spp.). These infected mammals were kept near prairie dogs that were later sold as pets.	[25]
2017	The largest West-African monkeypox outbreak began in September 2017 in Nigeria following very heavy rainfall and flooding. Active surveillance confirmed human monkeypox, and as of September 2019, a total of 176 human-monkeypox cases had been confirmed from 18 states.	[15,26]
2018	Four individuals traveling from Nigeria to the United Kingdom (UK) (*n* = 2; travel from Nigeria, bush meat possible for one case—Secondary Exposure during healthcare), Israel (*n* = 1; rodent carcasses in Nigeria), and Singapore (*n* = 1; travel to Nigeria and attended a wedding and eat bushmeat) became the first human-monkeypox cases exported from Africa and a related nosocomial-transmission event in the UK became the first confirmed human-to-human monkeypox transmission event outside of Africa. This explains the role travelers play in the spread of infectious-disease epidemics in new regions globally.	[27,28,29]
2019	A further case of human monkeypox was confirmed in the UK, again imported from Nigeria. Contact tracing was initiated and the smallpox vaccine (Imvanex) was procured. In 2019, a Nigerian travelling to Singapore for a training course developed skin lesions shortly after arrival and was diagnosed with human monkeypox.	[24,27]
2021	Reported human-monkeypox cases in the UK. Another case was reported in a returning traveler from Nigeria to Maryland, and another case in Texas in the USA.	[30]
2022	MPXV outbreak outside Africa starting from UK in early May 2022 MPXV outbreak in Africa in endemic and non-endemic African countries	[31]

**Table 2 vaccines-10-02091-t002:** The 2022 MPXV outbreak in Africa.

**African Countries That Have Not Historically Reported Monkeypox**	**Confirmed Cases**	**Deaths**
Egypt	1	0
Morocco	3	0
Sudan	18	1
Benin	3	0
Mozambique	1	1
South Africa	5	0
**African Countries that Have Historically Reported Monkeypox**	**Confirmed Cases**	**Deaths**
Liberia	3	0
Ghana	107	4
Nigeria	624	7
Cameroon	16	2
Central African Republic	12	0
Republic of the Congo	5	0
Democratic Republic of the Congo	206	0
Total as of 13 November 2022	1004	15 (CFR = 1.5%)
https://www.cdc.gov/poxvirus/monkeypox/response/2022/world-map.html, accessed on 13 November 2022		

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
