# Peer review of "Disease History, Pathogenesis, Diagnostics, and Therapeutics for Human Monkeypox Disease: A Comprehensive Review"

_vaccines, 2022, doi:10.3390/vaccines10122091_

Round 1
Reviewer 1 Report
The paper provides a comprehensive review of monkeypox. The paper is eligible for publication after minor revision.
1. The author can improve Table 1 by adding references for each event
2. The authors should update data on monkeypox on the day of resubmission
3. History of monkeypox is well presented
4. The authors should check for inconsistent abbreviations and non-abbreviation of words: monkeypox vs MPX, monkeypox virus vs MPXV, Democratic Republic of Congo vs DRC, VIG, WHO, etc.
5. There are several typographical mistakes. The authors should check
6. Diagnostic management: Consider merging these two paragraphs to avoid unnecessary repetitive use of those sentences.
Diagnosis of monkeypox starts with suspicion in cases suffering from pox-like lesions and a history of contact or travel to monkeypox-endemic regions. However, PCR is the preferred method to confirm the diagnosis. Whole-genome sequencing (WGS) is the gold standard for characterizing the monkeypox virus, but its use is restricted, particularly in developing countries. Serology and antigen detection are not advised due to the significant cross-reactivity with other Orthopoxviruses. Immunohistochemical (IHC) staining for orthopoxvirus antigens and visualization on electron microscopy (EM) are rarely used in clinical practice due to high technical skill and facility requirements.
Low blood urea nitrogen (BUN), increased transaminase levels, hypoalbuminemia, leukocytosis, and thrombocytopenia are some of the laboratory findings seen in monkeypox. The methods of serology and antigen detection are not advised because it is insufficiently specific due to the significant cross-reactivity with other Orthopoxviruses [73,74]. Additionally, immunohistochemical staining for orthopoxvirus antigens and visualiza-tion on electron microscopy are among diagnostic methods [75]; but rarely used in clinical practice due to high technical skill and facility requirements. Monkeypox culture-based testing shouldn't be done frequently in diagnostic or clinical laboratories [76].
7. Consider revising the therapeutics and vaccination section to enhance clarity. There are several discordant sentences and unnecessary sentence repetition and multiple typographical mistakes.
Examples:
It is available in the US as Jynneos, in Europe under the brand name Imvanex, and Imvamune in Canada [92]. Modified vaccinia Ankara (MVA) as a third-generation smallpox vaccine [93].
In the USA, the currently approved vaccines include ACAM2000, the purified clone of the Dryvax vaccine by Sanofi, and the modified vaccinia Ankara by Bavarian Nordic (JYNNEOSTM in the USA, IMVANEXTM in Europe, IMVAMUNETM in Canada). JYNNEOSTM is FDA-approved for both smallpox and monkeypox, and the USA Strategic National Stockpile contains both JYNNEOSTM and ACAM2000.
8. The authors should check the formatting for the citations throughout the manuscript

Author Response
|
Reviewer 1 - comments |
Response |
|
1. The author can improve Table 1 by adding references for each event
|
We are highly thankful for the suggestions: we have added the required references for each event in table 1. |
|
2. The authors should update data on monkeypox on the day of resubmission
|
We are highly thankful for the suggestions: the data has been updated as per the recent information. |
|
3. History of monkeypox is well presented
|
We are highly thankful for the appreciation. |
|
4. The authors should check for inconsistent abbreviations and non-abbreviation of words: monkeypox vs MPX, monkeypox virus vs MPXV, Democratic Republic of Congo vs DRC, VIG, WHO, etc. |
We are highly thankful for the suggestions. The abbreviations have been checked throughout the manuscript. All the abbreviations have been given at the end of the manuscript. |
|
5. There are several typographical mistakes. The authors should check |
We are highly thankful for the suggestions and have checked thoroughly.
|
|
6. Diagnostic management: Consider merging the two paragraphs to avoid unnecessary repetitive use of those sentences. |
We are highly thankful for such insightful suggestions; we have merged the repetitive information as per the suggestion and removed the unnecessary information.
|
|
7. Consider revising the therapeutics and vaccination section to enhance clarity. There are several discordant sentences and unnecessary sentence repetition and multiple typographical mistakes. Examples: It is available in the US as Jynneos, in Europe under the brand name Imvanex, and Imvamune in Canada [92]. Modified vaccinia Ankara (MVA) as a third-generation smallpox vaccine [93]. In the USA, the currently approved vaccines include ACAM2000, the purified clone of the Dryvax vaccine by Sanofi, and the modified vaccinia Ankara by Bavarian Nordic (JYNNEOSTM in the USA, IMVANEXTM in Europe, IMVAMUNETM in Canada). JYNNEOSTM is FDA-approved for both smallpox and monkeypox, and the USA Strategic National Stockpile contains both JYNNEOSTM and ACAM2000.
|
We have elaborated the section as per the suggestions of the both the reviewers.
|
|
8. The authors should check the formatting for the citations throughout the manuscript |
We are highly thankful for such insightful suggestions and revised the references section. |
Reviewer 2 Report
Saied et al. reviewed the history, pathogenesis, diagnostics, and therapeutics of MPXV. The manuscript is generally well-written and could be a good source of well-summarized information about Monkeypox. However, there are several points that the authors should address.
1. Table 1. All these details are not present in the reference [16]. I propose to insert a new column in the table to add the respective references.
2. Prevention and control: this needs to be expanded in more detail. For example i) Avoid touching oral and ocular rashes, letting other rashes dry, not covering the wound, or hydrating the rash with a moist cloth can help heal people from the infection and enhance the comfort of the affected person. Avoid direct touching the body surface, the crust of the rashes or pus, or droplet contact of sneezes or coughs of an infected person would help prevent the disease. ii) Surveillance and control plans in countries at high risk are highly recommended to detect and immediately respond to any suspected case of MPX. iii) Community engagement and rapid notification of the suspected case of MPXV to health authorities. iv) Increasing public health awareness about the virus and its transmission routes. v) Implementation of the One-Health approach during preparedness and response. Please add this part and citation: Parvin, et al. Ger. J. Microbiol. 2022.2(2): 1-15.
3. Introduction: Monkeypox is a rare zoonotic disease caused by a double-stranded DNA virus belonging to the Orthopoxvirus genus (MPXV), causing a smallpox-like disease in humans. Awkward: It should read: Monkeypox is a rare zoonotic disease caused by the Monkeypox virus (MPXV), a double-stranded DNA virus belonging to the Orthopoxvirus genus.
4. Figure 2. Please explain the figure to be stand-alone.
Please revise all abbreviations throughout the manuscript, considering expanding the full name when it first appeared in the paper
6. Please check the reference and make sure they followed the journal style

Author Response
|
Reviewer 2 - comments |
Response |
|
1. Table 1. All these details are not present in the reference [16]. I propose to insert a new column in the table to add the respective references.
|
We are highly thankful for the suggestions: we have added the required references for each event in table 1. |
|
2. Prevention and control: this needs to be expanded in more detail. For example i) Avoid touching oral and ocular rashes, letting other rashes dry, not covering the wound, or hydrating the rash with a moist cloth can help heal people from the infection and enhance the comfort of the affected person. Avoid direct touching the body surface, the crust of the rashes or pus, or droplet contact of sneezes or coughs of an infected person would help prevent the disease. ii) Surveillance and control plans in countries at high risk are highly recommended to detect and immediately respond to any suspected case of MPX. iii) Community engagement and rapid notification of the suspected case of MPXV to health authorities. iv) Increasing public health awareness about the virus and its transmission routes. v) Implementation of the One-Health approach during preparedness and response. Please add this part and citation: Parvin, et al. Ger. J. Microbiol. 2022.2(2): 1-15.
|
We are highly thankful for the suggestions: the information has been incorporated as per the suggestions. |
|
3. Figure 2. Please explain the figure to be stand-alone.
|
The figure legend has been explained in description. Additionally, all the abbreviations have been checked thoroughly. All the abbreviations have been given at the end of the manuscript for more readability to the readers. |
|
4. Introduction: Monkeypox is a rare zoonotic disease caused by a double-stranded DNA virus belonging to the Orthopoxvirus genus (MPXV), causing a smallpox-like disease in humans. Awkward: It should read: Monkeypox is a rare zoonotic disease caused by the Monkeypox virus (MPXV), a double-stranded DNA virus belonging to the Orthopoxvirus genus.
|
We are highly thankful for such insightful suggestions; The information has been reframed as per the suggestions.
|
|
8. Please check the reference and make sure they followed the journal style. MDPI and ACS Style Pires, C. Global Predictors of COVID-19 Vaccine Hesitancy: A Systematic Review. Vaccines 2022, 10, 1349. https://doi.o; rg/10.3390/vaccines10081349
|
We are highly thankful for such insightful suggestions, we revised references section. |
Round 2
Reviewer 2 Report
The manuscript is improved a lot. I recommend "Accept in present form"